# Multimodal Physiotherapist Intervention Program for Physical and Psychological Functioning in Children with Chronic Pain: Guiding Physiotherapy Intervention with the Pediatric Pain Screening Tool with Recommendations for Clinical Practice

**DOI:** 10.3390/jcm14113629

**Published:** 2025-05-22

**Authors:** Guillermo Ceniza-Bordallo, Javi Guerra-Armas, Mar Flores-Cortes, Sara Bermúdez-Ramirez

**Affiliations:** 1Department of Radiology, Rehabilitation and Physiotherapy, Faculty of Nursing, Physiotherapy and Podiatry, University Complutense of Madrid, 28040 Madrid, Spain; gcenizaprof@gmail.com; 2Experimental Health Psychology, Maastricht University, 6211 Maastricht, The Netherlands; 3Faculty of Health Sciences, University of Malaga, 29010 Malaga, Spain; marflco@hotmail.com; 4Clinimetry and Technological Development in Therapeutic Exercise Research Group (CLIDET), University of Valencia, 46010 Valencia, Spain; 5Rehabilitation Service, Hospital Dr. José Molina Orosa, Canary Service of Healthcare, 35500 Arrecife, Spain; sara-1997-mb@hotmail.com

**Keywords:** pediatrics, chronic pain, physical therapy modalities, multimodal therapy, risk-stratification, adolescents, physical activity, treatment

## Abstract

**Background/Objectives:** Pediatric chronic pain requires individualized care. The Pediatric Pain Screening Tool (PPST) allows for stratification of psychosocial and physical risk factors and may guide targeted interventions. However, its integration into multimodal physiotherapy programs remains unexplored. This exploratory feasibility case series study evaluated a PPST-guided, risk-stratified multimodal physiotherapy intervention in children aged 8–17 years with chronic pain. **Methods:** Participants were classified as low, medium, or high risk. Interventions were tailored accordingly. Outcomes were assessed pre- and post-intervention and included pain intensity, pain interference, psychological distress, and quality of life. **Results:** Ten participants (mean age = 13.5 years; 60% girls) were included. Six were classified as high, three as medium, and one as low risk based on the PPST. After an 8-week physiotherapist intervention program, pain interference significantly decreased (MD = −7.5; *p* = 0.040; d = 1.69), as did pain intensity at rest (MD = −3.1; *p* = 0.002; d = 2.60) and during movement (MD = −3.0; *p* = 0.004; d = 2.55), exceeding the MCID of 1.92. In the high-risk group, reductions were observed in anxiety (*p* = 0.006; d = 2.36), pain-related worries (*p* = 0.001; d = 3.79), fear of movement (*p* = 0.015; d = 1.62), and fear of pain (*p* = 0.002; d = 3.37). Eighty percent reported feeling “a great deal better” in the PGIC including all high-risk participants. **Conclusions:** These results supports the feasibility of integrating PPST risk stratification into multimodal management, providing a structured and effective framework for addressing pediatric chronic pain.

## 1. Introduction

Chronic pain in children and adolescents is a growing concern for healthcare systems, affecting approximately 20% of the pediatric population worldwide [1,2,3]. Beyond physical discomfort, chronic pain in childhood is associated with considerable emotional [4,5,6,7,8,9,10,11], social [12,13,14,15,16], and economic consequences [17,18,19,20,21,22,23,24,25], often leading to significant impairment in quality of life of both patients and their families [16,26,27,28,29,30]. Early and effective treatment is essential to prevent pain chronification, reduce disability, and improve long-term functional outcomes [31,32,33,34].

Multimodal and interdisciplinary approaches are currently recommended for the treatment of pediatric chronic pain, integrating physical, psychological, and educational strategies [35,36,37,38,39]. Within this framework, physiotherapy plays a key role, particularly through active interventions aimed to improve the tolerance to physical activity and exercise, reducing fear of pain, and restoring physical functioning [40,41,42,43,44,45]. However, implementing the right treatment at the right time remains a challenge, especially because of the absence of tools to identify which children might benefit most from specific interventions [39]. Stratified care consists of targeting treatments according to patient subgroups, aiming to enhance treatment benefits and minimize potential harms or unnecessary interventions [46]. While the stratified care approach in chronic musculoskeletal pain using tools such as the STarT MSK tool has exhibited meaningful clinical benefit such health-related cost benefits compared to usual care [47], some challenges have also been highlighted to better define the recommended treatment pathways [48,49]. Regarding these challenges, a wide range of stakeholders, including patients, parents, healthcare professionals and policy makers, should be involved in the development and implementation of stratified rehabilitation care [49]. In this context, the use of screening tools to stratify pediatric patients according to risk is gaining relevance [50,51,52,53,54], ultimately contributing to the advancement of precision medicine in the management of pediatric chronic pain [55,56].

The Pediatric Pain Screening Tool (PPST) [57] is a brief and validated measure that classifies children and adolescents with pain into low-, medium-, or high-risk groups based on physical and psychosocial factors. It has demonstrated its usefulness across different pain conditions—including musculoskeletal pain [50,51,52,57], headaches [53,58], sickle cell disease [54,59], and chronic postsurgical pain (CPSP) [60]. Previous studies using the PPST have focused primarily on predicting long-term outcomes and identifying children at risk of functional disability [57,58,59,60] to guide targeted interventions and inform clinical decision-making [50,51,52,53,54]. However, to date, no studies have evaluated its application in guiding physiotherapy treatment in pediatric populations with established chronic pain.

Risk stratification may be particularly useful in tailoring physiotherapy approaches based on each child’s clinical profile [35,50,51,52,53,54]. For example, while low-risk patients may benefit from brief educational or monodisciplinary interventions, children with higher levels of psychological distress or functional impairment may require more intensive, interdisciplinary approaches combining physiotherapy, psychological support, and cognitive-behavioral strategies [35]. As recommended by the Initiative on Methods, Measurement and Assessment of Pain in Clinical Trials (IMMPACT) and its recent pediatric update [55,61,62], individualized treatment should focus on the unique characteristics of each patient [63]. Merging risk-based stratification into a staged approach whereby treatment is targeted to subgroups of patients based on their responsiveness to complex, interdisciplinary interventions may provide an effective framework for real-world implementation and support a more precise and individualized approach in pediatric physiotherapy.

Despite this theoretical value, no previous studies have explored whether stratifying patients with the PPST [57] can guide and optimize physiotherapy treatment planning in routine clinical practice. Although the PPST [57] has primarily been used to predict long-term outcomes and identify risk of disability [57,58,59,60], its potential utility in guiding rehabilitation interventions—particularly physiotherapy—has not been studied. Understanding whether children at different risk levels respond differently to a standardized multimodal intervention could provide important insights into personalized care. Stratifying patients may allow clinicians to better understand clinical profiles, tailor the intensity and focus of interventions, and optimize outcomes within a common therapeutic framework [50,51,52,53,54]. Notwithstanding this potential relevance and its clinical implications, the role of risk-stratified multimodal physiotherapy in pediatric chronic pain has not been previously explored.

Given these considerations, the primary aim of this exploratory feasibility study was to explore the potential of PPST risk stratification following a multimodal physiotherapy intervention in children with chronic pain. Specifically, we aimed to explore changes in pain intensity, pain interference, anxiety, pain-related worries (often referred to as “pain catastrophizing” [64,65]), fear of pain, and quality of life. We hypothesized that children with chronic pain will reduce their pain intensity, pain interference, and fear of movement. Also, children will improve their psychological functioning and quality of life. The secondary aim of this study is to provide clinical recommendations for using this stratification framework to guide physiotherapy interventions in pediatric chronic pain.

## 2. Methods

### 2.1. Study Design

This study is a case series carried out from 1 January 2022 to 10 March 2023. This case series study follows Case Report (CARE) guidelines [66] supported by the Equator Network. The study received approval from the Ethics Committee of the Hospital 12 de October in Spain and adhered to all applicable regulations, including the Declaration of Helsinki, the General Data Protection Regulation (EU 2016/679), and the Law Enforcement Directive on Data Protection.

### 2.2. Participants

Children and adolescents aged between 8 and 17 years diagnosed with chronic pain (persistent pain at least during 3 months [67], including chronic musculoskeletal pain, chronic headache, chronic abdominal pain, chronic postsurgical pain) were recruited from private multidisciplinary pain clinic in the center of Spain. Inclusion criteria were (1) chronic pain affecting in daily life activities, (2) ability to complete self-reported questionnaires, and (3) parental consent and child assent. Exclusion criteria included (1) presence of neurological or genetic conditions affecting movement, (2) prior or ongoing physiotherapy interventions targeting pain, and (3) cognitive impairments that prevent questionnaire completion.

### 2.3. Recruitment

Participants were invited to take part in the study during their first clinical consultation. Those who agreed to participate provided informed consent (and assent for children up to 12 years old), after which all relevant study variables were collected, including demographic, clinical, and psychological measures.

### 2.4. Procedure

Initially, participants completed the PPST, which stratify patients into three risk categories based on their responses (into low-risk, moderate-risk, or high-risk groups). After completing the PPST, the participants completed all the baseline assessments prior to the first multimodal physiotherapy intervention (see Table 1 and Figure 1).

### 2.5. Intervention

The multimodal physiotherapy intervention consisted of 8 weeks of structured physiotherapy sessions tailored to each child’s PPST risk stratification. Initially, the patient’s condition was evaluated, and the main needs of the patient and the family were heard. After establishing the therapeutic goals, the treatment plan was designed in alignment with the patient’s objectives. The treatment was divided into 3 phases (mean of time 60 min), spread over 2 months divided into 3 phases. Sessions were conducted twice per week during 8 weeks by experienced physiotherapist specialized in pediatric pain management.

Symptom modification strategies: These techniques included massage, joint mobilization, and neural mobilizations [68,69,70,71]. Such interventions were aimed at temporarily reducing pain and improving function by directly addressing mechanical or neurophysiological contributors to the symptoms [68,70,71]. These strategies were adapted individually based on the patient’s clinical presentation and were integrated into the broader physiotherapeutic approach to optimize outcomes

Pain education: Pain neuroscience education focused on understanding pain mechanisms, fear of movement, and self-management strategies [41,42,72,73].

Exercise therapy: Progressive strengthening, mobility, and functional exercises adapted to each child’s tolerance and physical capabilities [74,75,76].

Cognitive-behavioral strategies: Techniques to reduce fear of movement and improve self-efficacy, such as gradual exposure and relaxation training [77,78,79].

Home exercise program: Patients received personalized exercise routines to reinforce therapy sessions and encourage active participation in daily activities [80,81].

### 2.6. Initial Phase

To modify symptoms (pain intensity, sensorial alterations), a physiotherapy intervention was performed combining manual therapy and neural mobilizations into regions affected. To improve the physical activity and the general self-perceived health, from the beginning of the treatment, weekly physical activity was prescribed. In addition, it is known to improve basal pain and promote proper treatment development as well as reduce anxiety, fear of movement, and fear-avoidance behaviors [82,83]. In the first sessions, it was observed that this activity was well tolerated, and it was gradually increased during the treatment.

An intervention on education in pain for patients and parents was also added. An adapted and easy to understand language was used using metaphors, stories, and images through adapted material recommended by specialists [41,73]. Clinical tool proposed by the literature such as videos, short stories, metaphors, and images were used and showed excellent results [42,73]. In all sessions, a space was left for questions and doubts that might arise during treatment.

### 2.7. Intermediate Phase

After observing during the first 3 sessions a positive evolution of the symptoms, adherence, and satisfaction with the treatment, new physiotherapy interventions were added to those that were already being developed.

Strength exercise (without reproduction of pain) began to be integrated using rubber bands and weights [84]. An intervention of graded exposure to movement with rubber bands and exercises was also included [84]. It began with accessory movement in joints and progressed toward the movement that produced the injury. During this intervention, pain evoked by movement and fear of pain were controlled. Some of these exercises were prescribed to perform at home. Increasing overall physical activity was prescribed.

### 2.8. Final Phase

In the last 2 sessions, pain symptoms were reduced, and the patient’s condition was very flattering. In the final assessment, the total resolution of the process and the clinically relevant improvement of pain, neural symptoms, and pain-related psychological factors were evidenced, and the patients were able to develop normal physical, school, and family activity. Patients’ global impressions of change were evaluated, and indications were given to continue with the prescribed sports practice.

### 2.9. Outcomes

Assessments were conducted at baseline (pre-intervention) and after completing the intervention (post-intervention). We collected demographic data age, birth gender, sex, race, ethnicity, and family economic income. The following validated outcome measures were used:

Pain risk stratification: The Pediatric Pain Screening Tool (PPST) [52] is a self-administered questionnaire designed as a child-friendly version of the nine-item tool. It includes two main components: physical and psychosocial factors. When completing the scale, patients are instructed to reflect on symptoms and emotions experienced over the past two weeks. Questions 1 through 8 require yes or no answers, which are scored as 0 or 1 accordingly. Item 9 asks about the overall impact of pain in the last two weeks, with response options ranging from “not at all” to “a whole lot”. These are scored as 0 for responses such as “not at all”, “a little”, and “some” and 1 for “a lot” and “a whole lot”. The total PPST score ranges from 0 to 9. The physical subscale, covering items 1 to 4, has a possible score between 0 and 4, while the psychosocial subscale, encompassing items 5 to 9, ranges from 0 to 5. A total score of 2 or lower indicates low risk, whereas scores of 3 or higher suggest a medium or high risk. Specifically, a psychosocial subscale score of 2 or below suggests medium risk, while a score of 3 or more on this subscale indicates high risk [57] (see Figure 2).

Pain intensity: Child pain intensity was assessed using the Numerical Rating Scale (NRS) [85]. The NRS is a self-report scale recommended for children aged 6 years and older. Children were asked, “How much pain have you felt in the past 3 months?” with response options ranging from 0 (“no pain”) to 10 (“the worst pain imaginable”). To interpret the clinical relevance of changes over time, the minimally clinically important difference (MCID) was set at −1.92 points on the NRS, based on the previous literature [86]. This threshold allowed us to determine whether reductions in pain were not only statistically significant but also meaningful from a clinical perspective.

Pain interference: Child pain inference was evaluated using the Pediatric Pain Interference Scale PROMIS (PROMIS-PPI) [87]. The PROMIS-PPI [87] is a self-reported eight-item scale that assesses the interference of pain in the daily life of the patient. The children use a Likert scale from “never” to “almost always” to rate the interference that pain produces in key components of daily life such as going to school, sleeping, or walking. T-scores were used in this study, where higher scores indicate higher levels of pain interference. The PROMIS-PPI demonstrated good reliability in a pediatric sample, with a Cronbach’s alpha of 0.90 [87].

Fear of movement: Child fear of movement was evaluated using the Tampa Scale for Fear of Movement-11 (TSK-11) [88]. The TSK-11 is a self-assessment scale composed of 11 items, through which fear of movement is evaluated. This scale is made up of 2 subscales, including somatic focus and avoidance of activity. The patients complete the instrument using a Likert scale of 4 responses ranging from “totally disagree” to “totally agree”. Higher scores indicate higher levels of fear of movement [88]. Total scores range from 11 to 44, with higher scores reflecting higher levels of fear of movement. TSK-11 showed good validity and reliability in pediatric samples [89]. To interpret the clinical relevance of changes over time, the MCID was set at −4.00 points on the TSK-11 based on the previous literature [90].

Anxiety: Child anxiety pain was assessed using the Children Pain Anxiety Symptoms Scale (CPASS) [91]. This instrument evaluates the extent to which children think, act, or feel in response to pain across four subscales: cognitive, physiological anxiety, fear, and escape/avoidance. The scale consists of 18 items rated on a six-point Likert scale, ranging from 0 (“never think, act, or feel that way”) to 5 (“always think, act, or feel that way”). Total scores range from 0 to 90, with higher scores reflecting higher levels of pain anxiety [91]. The CPASS demonstrated good reliability in a pediatric sample, with a Cronbach’s alpha of 0.88.

Pain-related worries: Child pain-related worries were assessed using the Pain Catastrophizing Scale-Children (PCS-C) [92]. The children rate each item using a Likert scale from 0 (“not at all”) to 4 (“extremely”), according to how they experience emotions in the situations that arise. Total scores range from 0 to 52, with higher scores reflecting higher levels of pain-related worries. The PCS-C demonstrated good reliability in pediatric samples, with a Cronbach’s alpha of 0.90.

Fear of pain: Child fear of pain was evaluated using the Fear of Pain Questionnaire-Children (FOPQ-C) scale [93]. The FOPQ-C is a 24-item self-report scale comprising two subdomains: “fear” and “avoidance”. Items are rated on a 5-point Likert scale ranging from 0 (“Totally disagree”) to 4 (“Totally agree”). Total scores range from 0 to 96, with higher scores reflecting greater levels of pain-related fear. The FOFQ-C demonstrated excellent reliability, with a Cronbach’s alpha of 0.92 in pediatric samples [93].

Health-related quality of life (HRQoL): Child HRQoL was assessed by PedsQL scale [94]. The PedsQL 4.0 is a self-report questionnaire composed of 23 items across four domains; “physical”, “emotional”, “social”, and “school functioning”. Respondents evaluate how often they experience certain difficulties using a 5-point Likert scale, with response options ranging from “*Never*” to “*Almost always*”. Scores are transformed to a 0–100 scale, where higher values reflect better perceived HRQoL [94].

Patient Global Impression of Change: The Patient Global Impression of Change (PGIC) [95,96] was used to capture participants’ subjective evaluation of changes in aspects such as activity limitations, symptom intensity, emotional well-being, and overall quality of life related to their pain experience. This measure consists of a 7-point Likert scale, with response options ranging from “no change” to “a great deal better”. While the PGIC is widely recommended as a core outcome indicator of overall treatment response in pediatric chronic pain trials [55], evidence regarding its psychometric properties in pediatric populations remains limited [97]. In this study, the PGIC was administered after the intervention.

### 2.10. Statistical Analysis

Demographic data were used to describe baseline characteristics of the sample through descriptive statistics, including means, standard deviations, and frequency distributions, depending on the type of variable. Similarly, baseline values of clinical variables were also presented using means and standard deviations to provide a detailed overview of participants’ clinical status at the start of the study. In addition, participants were categorized according to predefined risk groups, and data were presented based on this classification. To assess changes in the variables of interest between pre- and post-intervention, and differences between risk groups were conducted using Wilcoxon signed-rank tests, as the data did not meet the assumptions of normality and required a non-parametric approach, and post hoc analysis were conducted using Bonferroni tests. Effect sizes were calculated using Cohen’s d to assess the magnitude of pre- and post-intervention changes. Interpretation followed conventional thresholds: 0.2 (small), 0.5 (medium), and 0.8 (large) effect size [98]. Statistical significance was set at *p* < 0.05. All analyses were performed using IBM SPSS Statistics software, version 28.12. [99].

## 3. Results

A total of 10 children and adolescents aged between 8 and 17 years participated in this case series study. The mean age of participants was 13.5 years (SD = 2.4), with 60% identifying as girl at birth. All the participants were White and Hispanic/Latino (100%). Most participants (90%) came from families with a household income of <USD 60,000. Sociodemographic characteristics of participants are shown in Table 2.

The intervention was delivered by licensed physiotherapists with postgraduate training in pediatric pain with 5 years of clinical experience. A standardized protocol was used across therapists, developed collaboratively by the research team and discussed in regular supervision meetings. All participants completed the 8-week intervention program without any dropouts or missed sessions. A formal fidelity check conducted at the end of the intervention confirmed that the physiotherapy treatment was delivered as planned, with no modifications to the therapeutic content across participants. This ensured consistency in treatment delivery across the sample. None of the participants were taking amitriptyline or other tricyclic antidepressants during the intervention period.

At baseline (T1), risk stratification using the PPST classified six participants as high risk, three as medium risk, and one as low risk. The most commonly reported pain condition was low back pain (*n* = 5), followed by knee pain (*n* = 2), chronic headache (*n* = 1), and chronic postsurgical pain (*n* = 1). Mean pain intensity at rest across the sample was 6.3 (SD = 3.5), with higher values in the high-risk group (7.5, SD = 1.3), compared to the medium (5.7, SD = 2.3) and low-risk groups (4.0). Pain intensity during movement was also high, with a sample mean of 8.2 (SD = 0.5), ranging from 8.9 (SD = 1.0) in the high-risk group to 8.0 in the low-risk group. Pain interference, as measured using the PROMIS-PPI, was elevated in the overall sample (mean = 53.5, SD = 10.3), with progressively higher scores observed from the low-risk (45.0) to the high-risk group (55.4, SD = 7.8). The baseline characteristics for the different pain risk stratification groups are shown in Table 3.

Regarding psychosocial variables, participants showed moderate to high levels of fear of movement (mean = 20.5, SD = 5.2), anxiety (mean = 34.2, SD = 8.5), pain-related worries (mean = 30.5, SD = 6.5), and fear of pain (mean = 37.5, SD = 7.9). All psychosocial variables were markedly elevated in the high-risk group compared to the medium- and low-risk groups. HRQoL, was significantly impaired, with a mean score of 28.9 (SD = 5.6). Notably, HRQoL was lowest in the high-risk group (20.7, SD = 5.7) and highest in low-risk participants (34.0), indicating an important impact on quality of life. Bonferroni post-hoc analyses confirmed statistically significant differences between groups across nearly all variables (see Table 4).

After sixteen sessions of a multimodal physiotherapy intervention (T2), substantial clinical improvements were observed in pain intensity, physical function, and psychosocial outcomes across the entire sample. Resting pain intensity decreased 3.1 points on the NRS, representing both statistically (*p* < 0.002; Cohen d = 2.60) and clinically significant reductions. The most notable improvement was observed in the high-risk group, with a mean difference (MD) of 3.3 points (*p* < 0.003; Cohen d = 2.85). Similarly, pain during movement significantly decreased, with an average reduction of 3.0 points on the NRS-M (*p* < 0.004; Cohen d = 2.55), reaching the threshold for clinical relevance. The greatest reduction in movement-evoked pain was seen in the medium-risk group (MD = 3.4, *p* < 0.001; Cohen d = 2.90). These results show a MCID of pain intensity in movement-evoked pain [78] (see Table 5).

Pain interference showed a statistically and clinically significant reduction following the intervention, with a mean decrease of 7.5 points (*p* = 0.040; Cohen d = 1.69). Improvements were consistent across risk groups, with a reduction of 6.4 points in the high-risk group (*p* = 0.058; Cohen d = 0.89) and 7.0 points in the low-risk participant, both reflecting meaningful clinical change.

Functional and psychological recovery was also evident. Fear of movement levels decreased by 6.0 points in the low-risk group and by 7.2 points in the high-risk group, exceeded the MCID for the TSK-11 (≥4 points) [90], indicating clinically meaningful improvement [90]. Similar patterns of improvement were observed in anxiety (MD = −8.6, *p* < 0.059; Cohen d = 0.48), pain-related worries (MD= −8.5, *p* < 0.061; Cohen d = 0.40), and fear of pain (MD = −9.4, *p* < 0.018: Cohen d = 2.89), with more pronounced relative reductions in the lower-risk groups. These findings underscore the broad impact of the intervention on key psychological variables known to influence pain persistence and recovery.

HRQoL showed a statistically and clinically significant improvement following the intervention, with a mean increase of 7.3 points on the PedsQL scale (*p* < 0.077; Cohen d = 1.06). The largest improvement was observed in low-risk participants, with an increase of 11.0 points, clearly surpassing the threshold for clinically meaningful change. In the high-risk group, HRQoL improved by 9.3 points (*p* < 0.167; Cohen d = 1.66), also indicating a substantial clinical benefit.

Finally, according to the PGIC, the majority of participants (80%) reported feeling “a great deal better” after treatment. This perception of substantial improvement was most prominent in the high-risk group, where 100% of participants selected this response. In the medium-risk group, two participants (66%) reported being “a great deal better”, and one (33%) indicated feeling “better”. In contrast, the only participant in the low-risk group rated their change as “somewhat better”. No participants in any group reported “no change” or worsening of their condition (See Figure 3).

## 4. Discussion

This study aimed to explore changes in clinical variables in children and adolescents with chronic pain following a risk-stratified multimodal physiotherapy. After stratifying the participants as low, medium, and high risk, they were enrolled in a multimodal physiotherapy intervention combining manual therapy, physical activity. and education. Significant improvements in pain-related and psychosocial outcomes, including reductions in pain intensity (both at rest and during movement), pain interference, fear of movement, anxiety, pain-related worries, and fear of pain, were noted. Furthermore, 80% of the participants who received a multimodal physiotherapy intervention reported a higher response in the PGIC. The results showed a clinical improvement in pain-related functional and psychological outcomes within a multimodal physiotherapy intervention.

To our knowledge, this is the first study to report multidimensional outcomes after multimodal physiotherapist intervention in children and adolescents with chronic pain based on risk stratification using the PPST tool [57]. This system not only identifies patients who are in greater risk of experiencing a significant psychological and physical impact from their pain [52,54,57,60], it also provides valuable guidance for tailoring treatment according to individual needs [35,53,58].

Given that pediatric chronic pain is a highly individualized experience, with multiple biopsychosocial factors driving the prognosis for recovery and response to treatment, subcategorization is necessary for the personalized management. The use of the PPST in this context may offers clinicians a practical framework to stratify patients and inform treatment selection [35,53,58].

By applying this approach, our study is the first to show the potential clinical utility of integrating risk stratification into everyday practice. Stratifying patients into low-, medium-, and high-risk groups allowed us to understand better how different profiles of children and adolescents respond to treatment [35,53,58]. This approach made it possible to tailor the intervention to each group’s needs while still applying the same therapeutic components. Although all participants improved, the level and type of change varied between groups showing such individual differences. For example, children in the high-risk group showed greater reductions in resting pain and pain interference [100], while those in the low-risk group experienced more pronounced improvements in quality of life and emotional variables such as fear of movement and fear of pain. These findings suggest that early identification of risk level can help guide clinical decision-making and optimize outcomes by adjusting expectations and focus within the same intervention framework.

The multidisciplinary approach is the one that has shown the best results, and it is recommended by the scientific community [101]. Indeed, interdisciplinary multimodal pain treatments (IMPTs) are recommended as first-line strategies for pediatric chronic pain management [102]. In general, IMPT should be individualized to include one or more modalities, such as medication, rehabilitation strategies (e.g., physical therapy and/or occupational therapy), and psychological approach (e.g., cognitive-behavioral therapy, mindfulness-based stress reduction) [35,103]. Within this, physiotherapy has an essential role in IMPT [35]. The results show that including physiotherapy treatment in the multimodal approach offers great improvements in recovery time, reduces drug use, improves quality of life, increases tolerance to physical activity, and reduces pain and associated symptoms [101]. This is consistent with the results shown in this study where after a physiotherapy intervention, a very significant reduction in pain intensity, physical functioning, and psychological symptoms as well as greater HRQoL were evidenced. Chronic pain in the pediatric population is a multidimensional problem that requires all contributing dimensions to be assessed and prioritized. A call to incorporate personalized multimodal physiotherapy is thereby necessary to address its nature.

Part of the success of the physiotherapy treatment is that it is carried out intensively. It has been shown in clinical trials that this type of intervention is much more effective to both pain and disability reduction [101]. Furthermore, with this intervention model, it was possible to rapidly modify the symptoms. Thus, it rapidly influenced the painful experience and, therefore, the pain-related vulnerability factors that the patients presented.

For the modification of symptoms and short-term effects, a manual therapy intervention was performed combining massage therapy, manipulation, and mobilization. Manual therapy and massage therapy treatment have been shown to be highly effective in reducing pain intensity in the pediatric population [104]. High-speed techniques were included because they have been shown to be effective for disorders of the upper limbs in the adult population [105] in the clinic. It has been observed that it was also effective in children since they are capable of modulating pain through the release of endogenous substances [71]. Furthermore, manual therapy techniques have been shown to be capable of modulating pain by activating downward modulation [71].

On the other hand, one of the fundamental objectives of the treatment was to improve MEP and increase tolerance to exercise and physical activity. To do this, it began early with graduated exposure to movement [106]. In this way, exercises were carried out in the upper limbs without reproducing the pain. These types of interventions are known to be highly effective for injuries caused by exercise or sport [84] and have been shown to improve HRQoL, reduce fear of movement and fear of pain, and promote faster recovery [82,83,106]. Continuing in this line, it was proposed from the beginning to increase daily physical activity through aerobic exercise (bicycle, walking and elliptical), and the objective was to improve exercise tolerance and sleep hygiene, increase motivation, and reduce pain [82,83]. Additionally, using strength exercises, it was possible to improve the state of the muscles, increase self-efficacy, and enhance the overall state of health. In addition, the motivation of the patient greatly increased [82,83]. In this study, exercises were carried out in the treatment sessions. Then, when the patient had learned them, they had to be carried out at home twice a day. These indications were chosen because it gave the patient a tool to autonomously reduce the symptoms. In this way, the patients were directly included in their treatment. These methods managed to include the patient as an active part of the treatment, thereby increasing adherence to treatment and self-efficacy. Thus, the fear of movement was reduced very significantly, which helped the quick recovery and the confidence in the physiotherapy treatment.

In the final phase of the treatment, the fear-avoidance chronic pain model was reduced to non-clinical levels in all domains, so the patient regained full functionality. They could do high-intensity exercise without pain, and she got back the quality of life she had before the injury [107]. It is very important to recall that treatment satisfaction was excellent. It is an essential domain in pediatric chronic pain studies and a very important clinical information to evaluate treatment success [55].

## 5. Clinical Recommendations for Multimodal Physiotherapy Interventions According to Risk Stratification

Conventional therapeutic approaches in physiotherapy often rely on a one-size-fits-all strategy, failing to account for individuality, mechanisms, drivers, and a plethora of heterogeneous symptoms in people suffering from chronic pain [108]. To overcome these limitations, multimodal physiotherapy intervention has been proposed as a more adequate approach for the management of chronic pain [109,110] (see Figure 4).

Although there is currently no clear consensus on specific multicomponent approaches to pain management [111], multimodal physiotherapy intervention may integrate different modalities such as symptom modification strategies, education, movement representation techniques, movement-based active strategies, and sensorimotor re-training. The aim of these strategies is not only to reduce pain. These strategies also seek to improve neuroplasticity and physical functioning; reduce pain-related cognitions, emotions, and behaviors; and ultimately lessen disability and interference from chronic pain.

The risk stratification model based on the PPST tool may allow clinicians to classify patients according to their risk [35,53,58]. The diagram proposes a stepped-care approach that varies in intensity and complexity depending on the risk level: low, moderate, or high. As noted, precision pain medicine would consist of empirically based clinical decision algorithms that would identify appropriate interventions or combinations of treatments for specific pain patients (i.e., directing the right treatment, at the right dose, to the right patient, at the right time) [112]. Thus, risk stratification model based on the PPST brings us closer to the broad goal of physiotherapy intervention.

According to McCracken (2023) [113], a paradigm shift from a “evidence based protocol approach” to “process-based therapy (PBT)” is encouraged. PBT has been defined in the contextually specific use of evidence-based therapeutic processes linked to evidence-based therapeutic procedures for helping to solve the problems and promote the well-being of an individual [114]. To further understand the individual patient profiles and the unique presentation of a given diagnostic label, risk stratification identifies patients based on their pain experience and the impact of their disability to target personalized therapy. To this end, this framework is built on the premise of tailoring the intervention to the patient’s individual needs while maintaining a functional, patient-centered perspective [35,53,58]. At all levels, the intervention is anchored in pain science education and layered with additional components—such as physiotherapy, psychological support, medical treatment, occupational therapy, or even technology-based enhancers like virtual reality or biofeedback—based on case complexity [35,53,58]. A fundamental element across all levels is the collaborative setting of treatment goals with the child or adolescent, ensuring the intervention is meaningful, achievable, and aligned with their daily life and goal-valued priorities. This flexible yet structured model not only improves treatment efficiency but also supports more informed and personalized clinical decision-making.

For low-risk patients, the initial intervention can be delivered in a monodisciplinary outpatient format focused solely on pain science education. The goal is to provide clear, age-appropriate information about pain, reduce fear, promote self-efficacy, and maintain physical activity levels. Treatment planning should involve the patient setting short-term goals related to school attendance, hobbies, or daily movement, reinforcing autonomy and engagement. Positive indicators of progress include reductions in fear of movement, increased participation in daily activities, and improved body perception and movement confidence. However, if after a few sessions there is no meaningful improvement in function, coping, or physical activity engagement, clinicians should consider adding active physiotherapy or psychological support. In this group, it is essential to monitor for signs of persistent avoidance, ongoing fear of pain, or maladaptive beliefs about injury, as these may signal the need for a more comprehensive approach.

For moderate-risk patients, a more detailed assessment of underlying pain mechanisms is recommended to guide treatment planning. Depending on the clinical profile, a strengthened monodisciplinary intervention (e.g., education + physiotherapy or psychology) or a full multidisciplinary outpatient program may be appropriate. Physiotherapy should address motor and sensory impairments, include treatment of neural mechanosensitivity (e.g., neurodynamic mobilizations), and implement progressive exercises to increase movement tolerance and physical activity. Joint goal setting in this group should focus on regaining participation in meaningful activities, improving movement confidence, and reducing symptom interference in daily life. Key indicators of improvement include better range of motion, reduced sensory symptoms, increased daily physical activity, reduced fear of movement, and positive coping strategies. Moreover, persistent sensory disturbances, overprotective avoidance patterns, pronounced motor impairments, or poor treatment adherence may indicate the need to escalate care to a more intensive, coordinated intervention.

Flare-ups in pain during and after physiotherapy sessions, and particularly with exercise, has been reported to pose a significant barrier to adherence to multimodal physiotherapy programs [115,116]. Adverse effects during treatment have been an underreported problem, being a key contributor to low engagement in pediatric chronic pain. Compliance with multidisciplinary recommendations was reported to range from 46.7% to 100%, although the highest level of overall adherence was in the physiotherapy treatment [39]. While healthcare professionals may consider some adverse effects (e.g., increased pain intensity), to be part of the normal course or fluctuations of chronic musculoskeletal pain, adolescents in chronic pain and their parents have been highlighted as considering that the treatment may provoke an undesirable adverse effect such pain flare-ups, which then subsequently become an obstacle to continuing treatment [117]. It has been emphasized to provide an individualized treatment plan that manages pain flares, avoiding interference with the healthcare received by these pain patients [118]. Therefore, there is a need to incorporate pain control strategies into multimodal physiotherapy programs, especially in high- and moderate-risk groups. Although there is currently no consensus on the most appropriate methodology to use, some authors have suggested the “traffic light” framework to manage pain levels with different activities and behaviors and guide decision-making on pain variability during exercise-based interventions.

For high-risk patients, an intensive interdisciplinary approach is recommended, either through high-frequency outpatient care or partial hospitalization. This profile is typically characterized by high psychological burden, major functional impairment, and often significant sensory and motor disturbances. The treatment plan should be developed collaboratively with the patient and caregivers, addressing shared goals such as reducing pain intensity, returning to school or sports, or regaining social engagement. Physiotherapy should target sensory and motor alterations (e.g., hypo/hypersensitivity, muscle weakness, coordination deficits), address neuropathic pain components, and integrate graded exposure to movement. A structured plan to increase general physical activity (e.g., walking, cycling, aerobic exercise) should be included from the beginning, with close monitoring of progress. Indicators of improvement include normalization of movement patterns, reduced sensory symptoms, increased exercise tolerance, better sleep, and more active engagement in therapy. Conversely, lack of improvement despite intensive intervention, functional deterioration, increasing pain-related worries, or persistent extreme fear and avoidance should prompt a reassessment of therapeutic goals, consideration of enhanced psychological intervention, or the addition of advanced modalities such as virtual reality, biofeedback, or targeted medical treatments [119].

## 6. Limitations and Future Directions

This study presents several limitations that should be considered when interpreting the results. First, it is a case series with a small sample size, which limits the generalizability of the findings and the ability to perform robust statistical analyses. In addition, the low-risk group was represented by only one participant, preventing strong comparisons between risk levels. Moreover, the study lacked a control group and long-term follow-up, which makes it difficult to determine the sustainability of the observed effects or to attribute them exclusively to the physiotherapy intervention. Future studies should consider the design of a RCT with larger samples that are representative of the different risk profiles, ideally including medium- and long-term follow-up. Hence, future studies should further extend the generalizability of the results obtained in this exploratory study, including a wider cultural and socioeconomic homogeneity of the sample. Although randomized clinical trials (RCTs) are still best methodology in evaluating the effectiveness of interventions, several authors suggest that case series—along with other innovative designs such as N-of-1 trials or single-case experimental designs (SCEDs)—can also be appropriate [113,120,121,122,123]. These methodologies may help detect differences in treatment responses according to risk profiles [120], offering an unique opportunity to capture the individualized response to multimodal physiotherapy interventions and allowing for a more precise understanding of treatment efficacy at both intra- and inter-individual levels [113,122,123].

Another relevant limitation is the potential influence of uncontrolled external factors, such as family support, adherence to the home-based treatment, or concurrent participation in other interventions, all of which may have impacted the outcomes [100,124]. Although a risk-based stratification approach was used, differentiated treatment protocols were not systematically applied for each level, making it difficult to evaluate the specific impact of treatment personalization. It would also be valuable to develop and validate specific intervention protocols for each PPST-defined risk level, as well as to explore the mechanisms of change associated with observed improvements (e.g., neuroplasticity, emotional regulation, self-efficacy). In addition, future research could incorporate objective measures of physical function and motor activity and assess the feasibility of integrating complementary technologies such as virtual reality or biofeedback within a multimodal physiotherapy approach. Additionally, although MCID values were used to interpret the clinical relevance of changes in pain intensity and fear of movement, such thresholds are currently unavailable for other outcomes assessed in this study, including pain-related worries (PCS-P), anxiety (CPASS), pain interference (PROMIS-PPI), fear of pain (FOPQ-C), and quality of life (PedsQL). This limits the ability to draw firm conclusions regarding the clinical meaningfulness of changes observed in these domains. On the other hand, as the intervention was not blinded, outcomes relying on subjective recall—such as the PGIC—should be interpreted with caution, as they may be influenced by participant expectations or perceived improvements. Future studies should to replicate this study based on randomized control trial methods.

Finally, it will be important to include the patient perspective in the design of interventions with the use of participatory methodologies to ensure the relevance and clinical applicability of the programs developed. Using the PPST risk stratification approach, future RCTs can better tailor interventions to specific risk subgroups of patients to classify and recruit patients. Large-scale validation studies of this risk-based precision physiotherapy are also needed, whereas novel technologies such as artificial intelligence may play a differential role.

## 7. Conclusions

This study provides preliminary evidence on the effectiveness of a multimodal physiotherapy intervention in children and adolescents with chronic pain, adapted through a risk stratification model using the PPST tool. The results showed significant clinical improvements in pain intensity, functional outcomes, and various psychological variables, including fear of movement, pain-related worries, and quality of life. Further stratified RCTs are needed to confirm our preliminary results showing a promising role for risk stratification in physiotherapy in pediatric chronic pain. Stratified care made it possible to observe differences in treatment response according to individual risk profiles, supporting the need for personalized interventions based on patient characteristics. Furthermore, this work highlights the relevance of physiotherapy within a multimodal care model and emphasizes the clinical utility of screening tools such as the PPST to guide treatment decisions. A call for stakeholder engagement, including healthcare professionals, patients, and policy makers, is needed to underpin a successful real-world implementation of risk-stratified multimodal physiotherapy in care pathways. This framework is based on the premise of the patient-centered perspective, whereby active patient participation and physical functioning are key elements in the recovery process. Overall, these findings reinforce the high-value of integrating physiotherapy as a core component in the interdisciplinary multimodal pain treatment of pediatric chronic pain, within patient-centered, risk-informed, and functionally oriented care strategies.

## Figures and Tables

**Figure 1 jcm-14-03629-f001:**
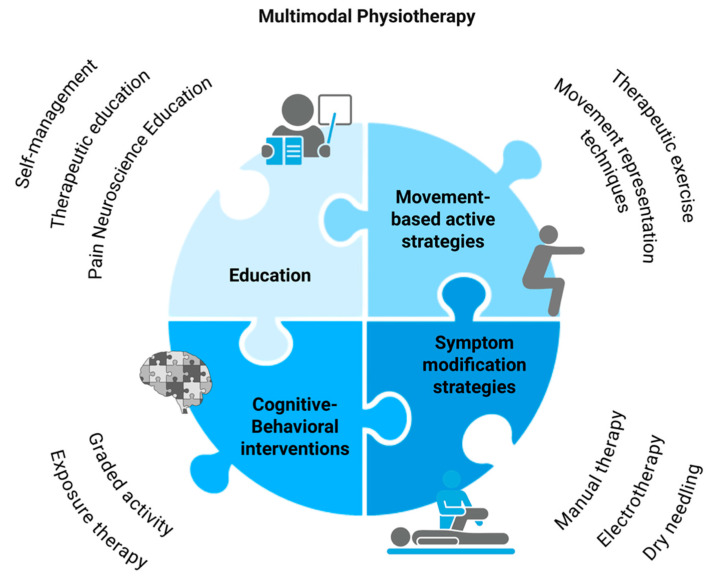
Components of multimodal physiotherapy intervention for pain management. This figure illustrates the four core components of a multimodal physiotherapy approach: (1) education, including self-management, therapeutic education, and pain neuroscience education; (2) movement-based active strategies, such as therapeutic exercise and movement representation techniques; (3) cognitive-behavioral interventions, including graded activity and exposure therapy; and (4) symptom modification strategies, such as manual therapy, electrotherapy, and dry needling. These components are integrated to address the multidimensional nature of pain and enhance treatment outcomes.

**Figure 2 jcm-14-03629-f002:**
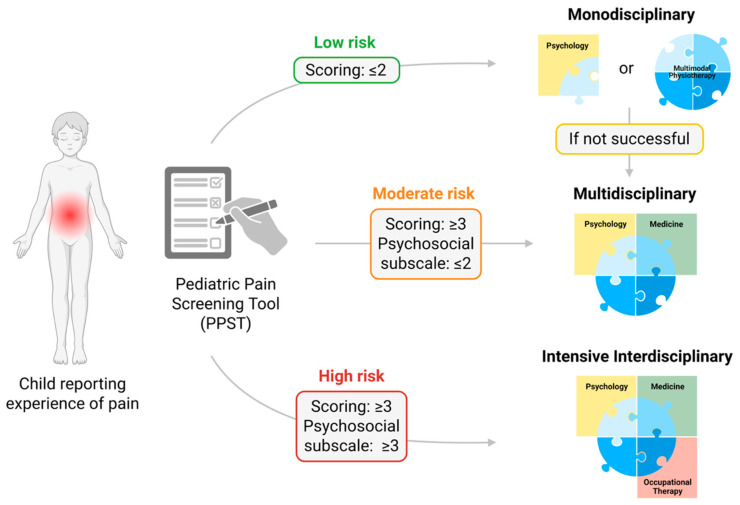
Risk stratification and treatment pathways based on the Pediatric Pain Screening Tool (PPST). The figure outlines a clinical decision-making model for children reporting pain using the PPST. Based on total and psychosocial subscale scores, patients are classified into low risk (score ≤2), moderate risk (score ≥ 3 and psychosocial subscale ≤ 2), or high risk (score ≥ 3 and psychosocial subscale ≥3). Low-risk patients may receive monodisciplinary treatment (e.g., psychology or multimodal physiotherapy). If not effective, referral to a multidisciplinary approach is recommended. Moderate-risk patients are directed to multidisciplinary care, typically involving psychology, medicine, and physiotherapy. High-risk patients are recommended for intensive interdisciplinary treatment, integrating psychology, medicine, physiotherapy, and occupational therapy.

**Figure 3 jcm-14-03629-f003:**
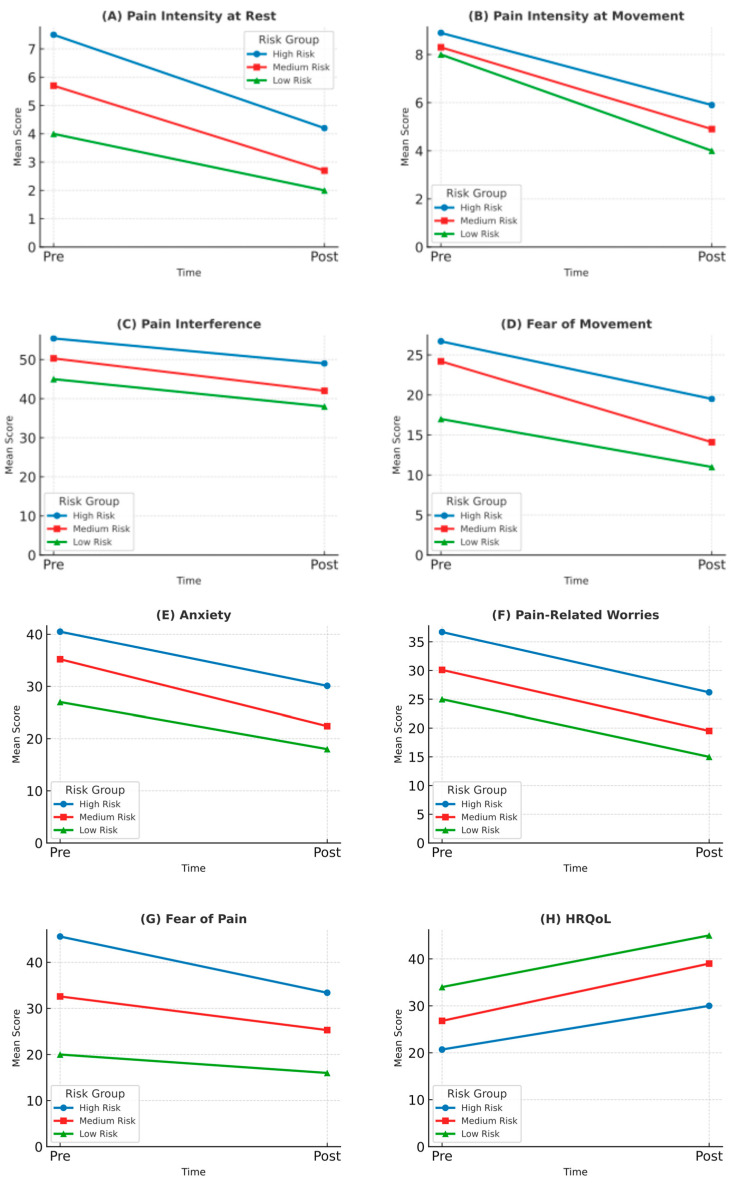
Pre- and post-intervention changes across physical and psychological outcomes by risk group. Panels illustrate mean scores at baseline (Pre) and after 8 weeks (Post) for each outcome variable, stratified by risk level (high, medium, and low) using the Pediatric Pain Screening Tool (PPST). (**A**) Pain intensity at rest (NRS); (**B**) Pain intensity during movement (NRS-M); (**C**) Pain interference (PROMIS-PPI); (**D**) Fear of movement (TSK-11); (**E**) Anxiety (CPASS); (**F**) Pain-related worries (PCS-C); (**G**) Fear of pain (FOPQ-C); (**H**) Health-related quality of life (PedsQL). Each line represents the average trajectory for a specific risk group: high (blue), medium (red), and low (green). All scores are reported as mean values. Visual trends suggest greater improvements in high-risk participants across most outcomes.

**Figure 4 jcm-14-03629-f004:**
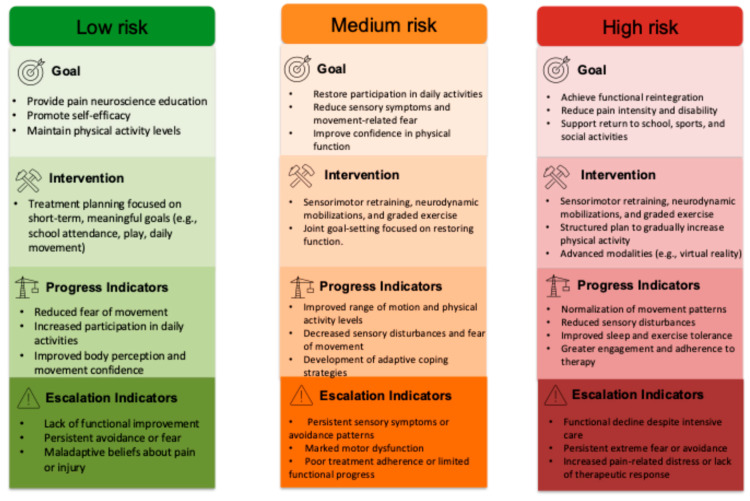
Clinical recommendations based on risk stratification using the Pediatric Pain Screening Tool (PPST). This figure presents a stratified clinical decision-making framework for guiding multimodal physiotherapy interventions in children and adolescents with chronic pain. Based on PPST scoring, patients are categorized into three risk levels—low, medium, and high—each associated with specific goals, recommended interventions, expected progress indicators, and escalation criteria.

**Table 1 jcm-14-03629-t001:** Physiotherapist treatment description and type of intervention by phases each visit.

Phase	Subjective Report	Treatment
	I. History and physical examination II. Pain and biopsychosocial factors	Initial evaluation Collaborative treatment goals planning (clinician–patient)
Initial phase	I. The movement evoked pain with definition of intensity and localization II. The patient’s ability to tolerate physical activity is evaluated. III. Fear of pain evaluation	Evaluation Pain-modifying strategies: manual therapy, neural mobilization Pain neuroscience education: (a) concept of pain, (b) differences between acute and chronic pain, (c) the process of pain chronification, (d) etiopathology of the child’s injury, (e) pain mechanisms and associated pain-related factors Physical activity: (a) education about the effects of physical activity, (b) prescription to gradually increase daily physical activity up to 4 h per week
Intermediate phase	I. Pain movement-evoked assessment II. Patient motivation was evaluated III. Fear of pain evaluation IV. Adverse events related to physical activity.	Evaluation Pain neuroscience education: (a) differences between pain and tissue damage, (b) the concept of the pain alarm system Physical activity: (a) addressing questions about physical activity and adverse events related, (b) graded movement exposure (c) prescription of daily physical activity (4 h per week)
Final phase	I. Final pain components assessment: pain intensity, pain movement evoked, psychological symptoms, HRQoL II. Plan for adaptation to real-life activities	Evaluation Pain neuroscience education: supporting the child in recognizing and adjusting to the variability of chronic pain symptoms, while also introducing tools for effective self-management Physical activity: (a) plan for adaptation to real-life activities, (b) address questions and fears of physical activity, (c) promote behaviors that support an active daily life

**Table 2 jcm-14-03629-t002:** Sociodemographic characteristics of participants included.

Variables	*n* (%)/Mean (SD)
Age	13.5 (2.4)
Sex	
Male	4 (40%)
Female	6 (60%)
Gender	
Boy	4 (40%)
Girl	6 (60%)
Race/Ethnicity	
White (Caucasian)	10 (100%)
Black (African American)	0 (0%)
Other	0 (0%)
Household income	
>100,000	0 (0%)
99,999–60,000	1 (10%)
59,999–40,000	6 (60%)
39,999–29,000	3 (30%)
<28,999	0 (0%)
Type of chronic pain	
Musculoskeletal chronic pain	7 (70%)
Low back pain	5
Knee	2
Chronic headache	2 (20%)
Chronic postsurgical pain	1 (10%)

SD = Standard deviation.

**Table 3 jcm-14-03629-t003:** Clinical variables at baseline (T1) in all samples and by risk group classification based on the PPST.

Variables	All Samples (*n* = 10) Mean/(SD)	High-Risk Group (*n* = 6) mean/(SD)	Medium-Risk Group (*n* = 3) Mean/(SD)	Low-Risk Group (*n* = 1) Mean/(SD)	Mean Differences *
Type of chronic pain					
Low back pain	5	3	2	1	–
Knee	2	1	1		–
Chronic headache	1	1			–
Chronic postsurgical pain	1	1			–
Pain intensity at rest (NRS)	6.3 (3.5)	7.5 (1.3)	5.7 (2.3)	4.0	a–b
Pain intensity at movement (NRS-M)	8.2 (0.5)	8.9 (1.0)	8.3 (0.2)	8.0	none
Pain interference (PROMIS-PPI)	53.5 (10.3)	55.4 (7.8)	50.3 (15.3)	45.0	a–b, b–c, a–c
Fear of movement (TSK-11)	20.5 (5.2)	26.7 (5.2)	24.2 (5.2)	17.0	a–c
Anxiety (CPASS)	34.2 (8.5)	40.5 (4.3)	35.2 (4.1)	27.0	a–b, b–c, a–c
Pain-related worries (PCS-C)	30.5 (6.5)	36.7 (2.4)	30.1 (4.5)	25.0	a–b, b–c, a–c
Fear of pain (FOPQ-C)	37.5 (7.9)	45.6 (3.2)	32.6 (3.2)	20.0	a–b, b–c, a–c
HRQoL (PedsQL)	28.9 (5.6)	20.7 (5.7)	26.8 (4.6)	34.0	a–c

SD = Standard deviation; NRS = Numerical Rating Scale; NRS-M = Numerical Rating Scale-Movement evoked; PROMIS-PPI = PROMIS-Pediatric Pain Interference; TSK-11 = Tampa Scale of Kinesiophobia; CPASS = Child Pain Anxiety Symptoms Scale; PCS-C = Pain Catastrophizing Scale-Children; FOPQ-C = Fear of Pain Questionnaire-Children; PedsQL = Pediatric Quality of Life; a = high-risk group; b = medium-risk group; c = low-risk group; * = post hoc analysis Bonferroni test 95%.

**Table 4 jcm-14-03629-t004:** Clinical variables at after treatment (T2) in all samples and based on risk group classification using PPST.

Variables	All Samples (*n* = 10) Mean/(SD)	High-Risk Group (*n* = 6) Mean/(SD)	Medium-Risk Group (*n* = 3) Mean/(SD)	Low-Risk Group (*n* = 1) Mean/(SD)	Mean Differences *
Pain intensity at rest (NRS)	3.2 (1.8)	4.2 (1.0)	2.7 (1.0)	2.0	a–b, b–c, a–c
Pain intensity at movement (NRS-M)	5.2 (0.8)	5.9 (0.7)	4.9 (0.5)	4.0	a–c
Pain interference (PROMIS-PPI)	46.0 (8.2)	49.0 (6.5)	42.0 (9.4)	38.0	a–b, b–c, a–c
Fear of movement (TSK-11)	16.2 (4.0)	19.5 (3.5)	14.1 (3.1)	11.00	a–c
Anxiety (CPASS)	25.6 (6.2)	30.1 (4.5)	22.4 (3.5)	18.0	a–b, b–c, a–c
Pain-related worries (PCS-C)	22.0 (5.3)	26.2 (3.1)	19.5 (3.5)	15.0	a–b, b–c, a–c
Fear of pain (FOPQ-C)	28.1 (6.7)	33.4 (4.0)	25.3 (3.5)	16.0	a–b, b–c, a–c
HRQoL (PedsQL)	36.2 (6.0)	30.0 (5.5)	39.0 (4.0)	45.0	a–c

SD = Standard deviation; NRS = Numerical Rating Scale; NRS-M = Numerical Rating Scale-Movement evoked; PROMIS-PPI = PROMIS Pediatric Pain Interference; TSK-11 = Tampa Scale of Kinesiophobia; CPASS = Child Pain Anxiety Symptoms Scale; PCS-C = Pain Catastrophizing Scale-Children; FOPQ-C = Fear of Pain Questionnaire-Children; PedsQL = Pediatric quality of life; a = high-risk group; b = medium-risk group; c = low-risk group; * = post hoc analysis Bonferroni test 95%.

**Table 5 jcm-14-03629-t005:** Changes in the clinical variables before (T1) and after treatment (T2) in all samples and based on risk group classification using the PPST.

Variables	All Samples (*n* = 10) Mean Differences (*p* Value/Cohen d)	High-Risk Group (*n* = 6) Mean Differences (*p* Value/Cohen d)	Medium-Risk Group (*n* = 3) Mean Differences (*p* Value/Cohen d)	Low-Risk Group (*n* = 1) Mean Differences	Mean Differences ^¶^
Pain intensity at rest (NRS)	−3.1 * (*p* = 0.002 d = 2.60)	−3.3 * (*p* = 0.003 d = 2.85)	−3.0 * (*p* = 0.006 d = 2.50)	−2.0 *	none
Pain intensity at movement (NRS-M)	−3.0 * (*p* = 0.004 d = 2.55)	−3.0 * (*p* = 0.006 d = 2.50)	−3.4 * (*p* = 0.001 d = 2.90)	−4.0 *	none
Pain interference (PROMIS-PPI)	−7.5 * (*p* = 0.040 d = 1.69)	−6.4 (*p* = 0.058 d = 0.89)	−8.3 * (*p* = 0.038 d = 1.09)	−7.0 *	none
Fear of movement (TSK-11)	−4.3 (*p* = 0.091 d = 0.26)	−7.2 * (*p* = 0.015 d = 1.62)	−10.1 * (*p* = 0.005 d = 1.95)	−6.0	b–c
Anxiety (CPASS)	−8.6 (*p* = 0.059 d = 0.48)	−10.4 * (*p* = 0.006 d = 2.36)	−12.8 * (*p* = 0.002 d = 2.76)	−9.0 *	b–c
Pain-related worries (PCS-C)	−8.5 (*p* = 0.061 d = 0.40)	−10.5 * (*p* = 0.001 d = 3.79)	−10.6 * (*p* = 0.001 d = 3.89)	−10.0 *	none
Fear of pain (FOPQ-C)	−9.4 * (*p* = 0.018 d = 2.89)	−12.2 * (*p* = 0.002 d = 3.37)	−7.3 (*p* = 0.068 d = 0.80)	−4.0	a–b, b–c, a–c
HRQoL (PedsQL)	7.3 (*p* = 0.077 d = 1.06)	9.3 (*p* = 0.167 d = 1.66)	12.2 * (*p* = 0.039 d = 2.66)	11.0 *	a–b, a–c
**Patient Global Impression of Change (PGIC)**	**All Samples** **(*n* = 10)** **N/Percentage**	**High-Risk Group** **(*n* = 6)** **N/Percentage**	**Medium-Risk Group** **(*n* = 3)** **N/Percentage**	**Low-Risk Group** **(*n* = 1)** **N/Percentage**	–
A great deal better	8 (80%)	6 (100%)	2 (66%)	0 (0%)	–
Better	1 (10%)	0 (0%)	1 (33%)	0 (0%)	–
Moderately better	0 (0%)	0 (0%)	0 (0%)	0 (0%)	–
Somewhat better	1 (10%)	0 (0%)	0 (0%)	1 (100%)	–
A little better	0 (0%)	0 (0%)	0 (0%)	0 (0%)	–
Almost the same	0 (0%)	0 (0%)	0 (0%)	0 (0%)	–
No change	0 (0%)	0 (0%)	0 (0%)	0 (0%)	–

NRS; Numerical Rating Scale; NRS-M; Numerical Rating Scale-Movement evoked; PROMIS-PPI; PROMIS Pediatric Pain Interference; TSK-11; Tampa Scale of Kinesiophobia; CPASS; Child Pain Anxiety Symptoms Scale; PCS-C; Pain Catastrophizing Scale-Children; FOPQ-C; Fear of Pain Questionnaire-Children; HRQoL = Health Related Quality of Life; PedsQL; pediatric quality of life; PGIC = Patient Global Impression of Change; a = high-risk group; b = medium-risk group; c = low-risk group; * = statistical differences in mean differences pre-post analysis (T1–T2), *p* < 0.05; ^¶^ = post hoc analysis Bonferroni test 95%.

## Data Availability

Deidentified individual participant data will be made available upon request that is accompanied by a justification for the request.

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
