# Peer review of "Multimodal Physiotherapist Intervention Program for Physical and Psychological Functioning in Children with Chronic Pain: Guiding Physiotherapy Intervention with the Pediatric Pain Screening Tool with Recommendations for Clinical Practice"

_jcm, 2025, doi:10.3390/jcm14113629_

Round 1
Reviewer 1 Report
Comments and Suggestions for Authors
Dear authors,
Thank you for the opportunity to review your manuscript entitled Multimodal physiotherapist intervention for physical and psychological functioning in children with chronic pain: insights from risk stratification with the Pediatric Pain Screening Tool and recommendation to clinical practice. This case series addresses an important and emerging topic in pediatric pain management and proposes a clinically relevant application of a risk stratification framework to tailor physiotherapy interventions. While the manuscript shows considerable potential, there are several methodological, interpretive, and conceptual areas that require clarification or improvement to meet the standards of a high-impact clinical journal.
Abstract
The abstract is informative and provides a summary of the study’s aim, design, and key results. However, it conflates statistical and clinical significance without providing exact p-values or the thresholds for meaningful change across outcomes. Additionally, terms like “notably” and "particularly pronounced" are too subjective and should be replaced by quantitative descriptions (e.g., effect sizes, MCID attainment). Clarify whether this was an exploratory feasibility study or a hypothesis-driven evaluation.
Introduction
The introduction presents a well-articulated rationale for adopting risk stratification in pediatric chronic pain. The description of the Pediatric Pain Screening Tool (PPST) is clear, and the literature cited supports its utility in various clinical contexts. However, the introduction would benefit from a more critical synthesis of prior research. For instance, while the use of PPST in identifying functional risk is cited, its application for guiding physiotherapy specifically is not adequately contextualized. The novelty of applying risk stratification to physiotherapy should be supported by more discussion of existing stratified care models in rehabilitation (e.g., STarT Back) and why they have or have not been implemented in pediatrics.
Methods
The case series design is appropriate for this type of exploratory clinical investigation. Nonetheless, the methodological reporting should be strengthened in several ways:
- The criteria for defining “clinical relevance” of change across outcomes are inconsistently reported. While MCIDs are given for pain intensity, no such benchmarks are described for PROMIS-PPI, PCS-C, or other psychological measures. Include them explicitly or justify their absence.
- The stratification into low, medium, and high risk using the PPST is well defined, but the cutoffs used for the psychosocial subscale require clearer validation reference.
- The intervention content is extensive and clearly described, but it is not evident whether the delivery was systematically different across risk groups. While the intervention is tailored, the same components (manual therapy, education, graded exposure, etc.) appear to have been used across all groups. Clarify what, if any, differences in dose or focus were implemented by risk level.
- There is no mention of therapist fidelity, adherence, or supervision, which weakens the interpretability of intervention consistency.
- The use of Wilcoxon signed-rank tests and Bonferroni corrections is appropriate, but given the small sample, the authors should report effect sizes (e.g., r or matched-pairs rank biserial correlation) to better interpret the magnitude of change.
- The rationale for categorizing a single low-risk participant should be addressed. Statistical comparison is not feasible with one subject, and inclusion of these data may be misleading.
Results
The results section is well structured and supported by comprehensive tables. Improvements in pain, interference, and psychological functioning are clearly presented. However, a few issues need clarification:
- There is an overemphasis on statistical significance without adequate discussion of clinical significance. Some mean differences are small, and readers should be informed whether they meet established MCIDs.
- The risk group comparisons are interpreted as if powered analyses were performed. Given the sample size (n=10), these comparisons should be framed as exploratory and descriptive only.
- PGIC responses are encouraging, but the limitations of subjective recall in non-blinded interventions must be acknowledged.
- Data visualization would strengthen the reader's understanding. Consider adding line graphs or boxplots to show change over time by group.
Discussion
The discussion addresses the key findings and interprets them within a clinical context. However, it occasionally overstates the implications. For instance, the claim that stratification “allowed us to tailor the intervention” is not entirely supported by the methods, which suggest a largely uniform protocol with some adaptations. Similarly, the assertion that PPST stratification supports “precision physiotherapy” is compelling but would be more appropriate as a hypothesis for future validation studies rather than a conclusion drawn from a small case series. The discussion should also:
- Address the potential for regression to the mean, especially in high baseline symptom groups.
- More clearly discuss the limits of generalizability, including the cultural and socioeconomic homogeneity of the sample.
- Acknowledge the influence of parental involvement and family environment on outcomes, especially given the developmental age of participants.
- Expand on how this study informs future trial design (e.g., stratified RCTs, stepped care models).
Conclusion
The conclusion accurately summarizes the study’s contribution but should avoid definitive claims. Phrases such as “provides preliminary evidence” or “suggests potential utility” are more appropriate than “demonstrates effectiveness.” Reframe the takeaway messages to emphasize the promise of risk stratification in physiotherapy planning, with a call for further controlled research.
Author Response
Comment 1- Abstract: The abstract is informative and provides a summary of the study’s aim, design, and key results. However, it conflates statistical and clinical significance without providing exact p-values or the thresholds for meaningful change across outcomes. Additionally, terms like “notably” and "particularly pronounced" are too subjective and should be replaced by quantitative descriptions (e.g., effect sizes, MCID attainment). Clarify whether this was an exploratory feasibility study or a hypothesis-driven evaluation.
Response: Thank you very much for your suggestions. We have clarified that this is an exploratory feasibility study. We included quantitative descriptions and adopted more precise and objective language in the results section. The abstract has been fully restructured in accordance with your comments. A 244-word abstract has been implemented, including a brief introduction, identification of the gap in the literature, study objective, methods, results, and conclusions.
Comment 2- Introduction: The introduction presents a well-articulated rationale for adopting risk stratification in pediatric chronic pain. The description of the Pediatric Pain Screening Tool (PPST) is clear, and the literature cited supports its utility in various clinical contexts. However, the introduction would benefit from a more critical synthesis of prior research. For instance, while the use of PPST in identifying functional risk is cited, its application for guiding physiotherapy specifically is not adequately contextualized. The novelty of applying risk stratification to physiotherapy should be supported by more discussion of existing stratified care models in rehabilitation (e.g., STarT Back) and why they have or have not been implemented in pediatrics.
Response: Thank you very much for your suggestion. Further details on stratified care models in rehabilitation have been included in the introduction.
Comment 3- Methods: The case series design is appropriate for this type of exploratory clinical investigation. Nonetheless, the methodological reporting should be strengthened in several ways:
- Comment 3.1. The criteria for defining “clinical relevance” of change across outcomes are inconsistently reported. While MCIDs are given for pain intensity, no such benchmarks are described for PROMIS-PPI, PCS-C, or other psychological measures. Include them explicitly or justify their absence.
Response: Thank you for your observation. We agree that the inclusion of MCID values is essential to interpret the clinical relevance of outcomes. In the current version of the manuscript, we have incorporated MCID thresholds for pain intensity and fear of movement (TSK-11), which are supported by existing literature in pediatric populations. However, to the best of our knowledge, MCID have not yet been established in children or adolescents for the measures used to assess pain-related worries (PCS-P), anxiety (CPASS), pain interference (PROMIS-PPI), fear of pain (FOPQ-C), or health-related quality of life (PedsQL). Therefore, we were not able to include MCID interpretations for these outcomes. We acknowledge this as a limitation and have now clarified it in the limitation section.
- Comment 3.2: The stratification into low, medium, and high risk using the PPST is well defined, but the cutoffs used for the psychosocial subscale require clearer validation reference.
Response: Thank you for your comment. We agree that the cutoffs used for the psychosocial subscale of the PPST require appropriate referencing. We have now clarified this in the manuscript by citing the original study by Simons et al. (2010), which established the ≥3 cutoff for identifying elevated psychosocial risk. This reference has been added to the section describing the stratification criteria to support the validity of the thresholds used. Additionally, we incorporated a figure (figure 2) to help undertanding the tree-decition-making. We hope all the changes helps to improve the manuscript.
- Comment 3.3: The intervention content is extensive and clearly described, but it is not evident whether the delivery was systematically different across risk groups. While the intervention is tailored, the same components (manual therapy, education, graded exposure, etc.) appear to have been used across all groups. Clarify what, if any, differences in dose or focus were implemented by risk level.
- Response: Thank you for your insightful comment. While the intervention was guided by risk stratification using the PPST, we intentionally ensured that the core physiotherapy components—such as education, manual therapy, graded exposure, and exercise—were delivered similarly across risk groups. This approach was adopted to maintain consistency and allow for meaningful comparison of outcomes across subgroups. The main differences by risk level were related to the involvement of additional professionals (e.g., psychologists, pain specialists) and the degree of interdisciplinary coordination, particularly in the high-risk group. We have clarified this point in the revised methods section.
- Comment 3.4: There is no mention of therapist fidelity, adherence, or supervision, which weakens the interpretability of intervention consistency.
- Response: Thank you for this important observation. Therapist fidelity and participant adherence were formally assessed at the end of the 8-week intervention. All participants completed the full duration of the program and attended all scheduled sessions. Additionally, no deviations or changes were made to the planned therapeutic content across participants. Interventions were delivered by experienced pediatric physiotherapists following a standardized protocol, and regular supervision meetings were conducted to ensure consistency in delivery. These details have now been clarified in the results section.
- Comment 3.5: The use of Wilcoxon signed-rank tests and Bonferroni corrections is appropriate, but given the small sample, the authors should report effect sizes (e.g., r or matched-pairs rank biserial correlation) to better interpret the magnitude of change.
- Response: Thank you for this helpful suggestion. In response to your comment, we have included effect sizes for all variables assessed in the study to better evaluate the magnitude of change between pre- and post-intervention assessments. This was done to enhance the interpretability of the results, particularly given the small sample size. Effect sizes are now reported alongside the corresponding statistical tests throughout the results section, Table 5 and in the abstract.
- Comment 3.6: The rationale for categorizing a single low-risk participant should be addressed. Statistical comparison is not feasible with one subject, and inclusion of these data may be misleading.
- Response: Thank you for this important observation. As noted in the manuscript, the low-risk group consisted of a single participant. This has been clearly and transparently indicated in both the results section and the corresponding tables. Furthermore, no statistical comparisons were made between risk groups. Instead, we analyzed pre- and post-intervention values within each group without comparing the magnitude of change across groups. This approach avoids potential bias and ensures that the inclusion of the low-risk participant does not influence the interpretation of group-level outcomes. Our intention was to report the data objectively and comprehensively, including all participants enrolled in the study.
Comment 4- Results: The results section is well structured and supported by comprehensive tables. Improvements in pain, interference, and psychological functioning are clearly presented. However, a few issues need clarification:
- Comment 4.1: There is an overemphasis on statistical significance without adequate discussion of clinical significance. Some mean differences are small, and readers should be informed whether they meet established MCIDs
- Response: Thank you for this valuable comment. In the revised manuscript, we have placed greater emphasis on the interpretation of clinical significance by indicating whether the observed changes met established minimal clinically important differences (MCIDs). Specifically, we now highlight that the changes in pain intensity and fear of movement exceeded known MCID thresholds in pediatric populations. For other variables—such as pain-related worries, anxiety, pain interference, fear of pain, and quality of life—MCID thresholds have not yet been established in children or adolescents. We have acknowledged this limitation in the discussion section and avoided overinterpreting statistically significant results in these domains.To further address your concern and avoid overemphasis on statistical significance, we have also included effect sizes (Cohen’s d) for all outcome measures. This allows for a clearer and more transparent understanding of the magnitude and clinical relevance of the changes observed.
- Comment 4.2: The risk group comparisons are interpreted as if powered analyses were performed. Given the sample size (n=10), these comparisons should be framed as exploratory and descriptive only.
- Response: Thank you very much for this insightful comment, which has significantly contributed to improving the clarity of the manuscript. We fully agree that, given the small sample size (n = 10), the comparisons across risk groups should be interpreted as exploratory and descriptive rather than inferential. We have now made this explicit in the description of the study design and have also clarified it in the abstract. This change ensures that the findings are appropriately framed within the scope and limitations of this feasibility study.
- Comment 4.3: PGIC responses are encouraging, but the limitations of subjective recall in non-blinded interventions must be acknowledged.
- Response: Thank you for this thoughtful comment. We agree that, in the context of a non-blinded intervention, subjective measures such as the PGIC are susceptible to recall bias and expectancy effects. We have now acknowledged this limitation in the limitation section to provide a more balanced interpretation of the findings.
- Comment 4.4: Data visualization would strengthen the reader's understanding. Consider adding line graphs or boxplots to show change over time by group.
- Response: Thank you for this thoughtful comment. The line graphs has been included in the manuscript (Figure 3).
Comment 5- Discussion: The discussion addresses the key findings and interprets them within a clinical context. However, it occasionally overstates the implications. For instance, the claim that stratification “allowed us to tailor the intervention” is not entirely supported by the methods, which suggest a largely uniform protocol with some adaptations. Similarly, the assertion that PPST stratification supports “precision physiotherapy” is compelling but would be more appropriate as a hypothesis for future validation studies rather than a conclusion drawn from a small case series. The discussion should also:
- Address the potential for regression to the mean, especially in high baseline symptom groups.
- More clearly discuss the limits of generalizability, including the cultural and socioeconomic homogeneity of the sample.
- Acknowledge the influence of parental involvement and family environment on outcomes, especially given the developmental age of participants.
- Expand on how this study informs future trial design (e.g., stratified RCTs, stepped care models).
Response 5: Thank you for these thoughtful and constructive comments. We have carefully revised the discussion section to ensure that the implications of our findings are presented in a balanced and accurate manner. Specifically, we have modified the language around the use of stratification to avoid overstating its impact, clarifying that the intervention protocol was largely uniform with some adaptations by risk level. Additionally, we now frame the potential of PPST stratification to support precision physiotherapy as a promising direction for future validation studies rather than a definitive conclusion. These additions strengthen the discussion and improve the interpretability and relevance of our findings.
Comment 6-Conclusion: The conclusion accurately summarizes the study’s contribution but should avoid definitive claims. Phrases such as “provides preliminary evidence” or “suggests potential utility” are more appropriate than “demonstrates effectiveness.” Reframe the takeaway messages to emphasize the promise of risk stratification in physiotherapy planning, with a call for further controlled research.
Response: Thank you for this thoughtful observation. We fully agree on the importance of avoiding overstatements, especially in exploratory research. We have carefully reviewed the conclusion and confirmed that it maintains a cautious tone, aligned with the preliminary and descriptive nature of this case series. No definitive claims regarding effectiveness are made. Instead, the conclusion emphasizes the potential utility of PPST-guided stratification to inform physiotherapy planning and highlights the need for further controlled research. Therefore, we believe the current framing appropriately reflects the study's scope and level of evidence.
Reviewer 2 Report
Comments and Suggestions for Authors
Very good review of using PpST to stratify chronic pain kids. Our clinic at McW has been using this tool to determine high risk- MD and psychologist v low risk NP alone. A few suggestions to make it better
p3 brief list of pain diagnoses in participants section
p4 often hard to prove patients do home pT. Was CbT also part of the treatment?
all good tools, PPSt, NRS,PROMS,PCS
Were meds used like amitriptyline
really wish there were more than just ten patients
Comments on the Quality of English LanguageEnglish was fine
Author Response
Comment 1- Very good review of using PpST to stratify chronic pain kids. Our clinic at McW has been using this tool to determine high risk- MD and psychologist v low risk NP alone. A few suggestions to make it better.
Response: We appreciate all your comments. We are very excited that you are indeed already using this tool in your clinical practice.
Comment 2- p3 brief list of pain diagnoses in participant’s section
Response: Thank you for your suggestion. We have now included a brief list of the pain diagnoses observed in our sample in the participants section, to provide additional clinical context.
Comment 3- p4 often hard to prove patients do home pT. Was CbT also part of the treatment?
Response: This study has focused specifically on the physical therapy approach, but within the Interdisciplinary multimodal pain treatment, psychological strategies such as CBT are recommended.
Comment 4- All good tools, PPSt, NRS,PROMS,PCS
Response: Thank you very much. We consider that these PROMs provide great value to the evaluation of the pediatric patient.
Comment 5- Were meds used like amitriptyline
Response: Thank you for your question. None of the participants in this study were taking amitriptyline at the time of the intervention. We just clarified this point into results section.
Comment 6- really wish there were more than just ten patients
Response: A larger sample size is ideally needed. However, given that this study was the first to utilize risk-based stratification in pediatric chronic pain, we have conducted an exploratory study that opens up opportunities for future studies with larger sample sizes.
Reviewer 3 Report
Comments and Suggestions for Authors
23 April 2025
The decision on the manuscript, titled ‘Multimodal physiotherapist intervention for physical and psychological functioning in children with chronic pain: in-sights from risk stratification with the Pediatric Pain Screening Tool and recommendation to clinical practice’ by Ceniza-Bordallo M, submitted to Journal of Clinical Medicine
Dear Authors,
Chronic pain affects approximately 20% of children globally, often impairing their physical function, emotional well-being, and overall quality of life. Despite recommendations for multimodal care, clinicians still face challenges in personalizing physiotherapy due to the lack of validated tools to guide treatment selection. In the current manuscript entitled ‘Multimodal physiotherapist intervention for physical and psychological functioning in children with chronic pain: in-sights from risk stratification with the Pediatric Pain Screening Tool and recommendation to clinical practice,’ Ceniza-Bordallo and colleagues. investigate investigates the effectiveness of a multimodal physiotherapy intervention tailored by Pediatric Pain Screening Tool (PPST) risk stratification to improve physical and psychological outcomes in children with chronic pain.
A key strength of this study lies in its innovative use of the PPST to tailor physiotherapy interventions based on individual risk profiles. By integrating physical, educational, and psychological components, the approach mirrors real-world clinical practice. It’s practical, structured, and responsive to each child’s needs. The clear improvements across physical and emotional outcomes suggest this method holds great promise for personalized pediatric pain management.
This study addresses an important and timely topic with intriguing findings that will capture the interest of Journal of Clinical Medicine readers. While the arguments are persuasive, they would be stronger with additional evidence and more detailed analysis. Responding thoroughly to the reviewers' comments and refining these areas will elevate the manuscript to a publishable standard.
Comments:
Title: Provide a concise and informative title that accurately reflects the key message of this study, as this is the most essential aspect of the manuscript. Suggestions: Stratify, Specify, Satisfy: Multimodal Physiotherapy for Pediatric Chronic Pain; Precision in Motion, Protection from Emotion: Risk-Guided Therapy in Youth Pain; From Screening to Strength: Tailored Physiotherapy for Pediatric Pain Relief.
Abstract: Please structure the abstract by summarizing the background, methods, results, and conclusion in a well-balanced 200-word passage, without using section headings. Begin with a concise general introduction (1–2 sentences), followed by a more specific context (2–3 sentences) that leads into the research problem. Transition smoothly into the study’s objectives. When presenting results, focus on the key findings clearly and concisely, ending with a sentence that connects them to the wider scientific discussion. The conclusion should open with a strong statement, such as "This study demonstrates," emphasizing its main contribution. Wrap up with 2–3 sentences that provide a broader perspective, making the findings relevant and engaging for a diverse scientific audience. Avoid presenting statistical details in the abstract.
Keywords: Make sure the majority of keywords are from Medical Subject Headings (MeSH) and use as many keywords as possible in the title and the first two sentences of the abstract.
Graphical abstract: I highly recommend presenting an informative graphical or video abstract.
Introduction: Expand this section to improve clarity and reader engagement. Begin with a broad context to set the stage, then gradually narrow the focus to the specific issue being examined. Organize the introduction into well-structured paragraphs, aiming for a total of around 1000 words to thoroughly develop key concepts. Ensure accessibility for readers from different disciplines by clearly explaining the study’s purpose. Move logically from general background to a precise statement of the research objectives. Highlight the study’s significance by emphasizing how it addresses gaps in the field. Maintain a smooth, coherent flow to guide the reader seamlessly into the next sections, making the introduction both comprehensive and compelling. Avoid numbering in the text.
Methods: To enhance clarity and reproducibility, detail the therapists’ qualifications and standardization procedures used across sessions. Including a session fidelity checklist or protocol adherence rating would bolster methodological rigor. Short, structured descriptions of each intervention phase can also aid replication and provide clearer insight into clinical decision-making processes. Avoid using bullets; rather, consider presenting tables.
Results: Consider integrating visual summaries such as bar graphs or line charts to illustrate key pre- and post-intervention changes. These visuals can enhance reader comprehension. Additionally, briefly highlighting effect sizes or clinical significance alongside p-values would deepen interpretation. Keep numerical results concise while emphasizing the most meaningful trends across risk groups. Avoid including statistical details within the main text and instead direct readers to the tables for reference. To enhance clarity, wrap up this section with a concise summary that highlights the key findings. A well-structured final paragraph will provide a clear overview of the results while emphasizing their importance in the broader scope of the study.
Discussion: This section is well written. Please refer to the following general guidelines for crafting a discussion section. I recommend not presenting subheadings in the discussion. Present the discussion as a unified narrative, avoiding subsections, and organize it into several paragraphs totaling around 1,500 words. Start with an opening paragraph that introduces the discussion, and end with a concise summary that ties back to the main findings from the results. Use this section to build arguments that address the study’s primary objectives, key challenges, and the advancements in knowledge or tools needed to tackle them. Place the findings within a broader field perspective, emphasizing the significance of this research and showcasing how it expands on prior work while offering new insights. Discuss the theoretical and practical implications, as well as how the study can inspire future research. To enhance the discussion, evaluate the strengths and limitations of the study and reflect on its relevance to clinical applications, highlighting the practical impact of the findings. This structure will ensure the discussion is balanced, engaging, and meaningful.
Conclusion: This section is also well written. Just for the sake of improvement, I would appreciate it if the authors could benefit from the following general guidelines: The conclusion section should be thoughtfully crafted to deepen the reader’s understanding of the study’s significance with approximately 150–200 words. Ideally, this section should begin with a concise summary of the authors’ key message, followed by a clear presentation of the key research findings. Authors should then contextualize these results by discussing their broader theoretical and practical implications, emphasizing how the study advances knowledge in the field. Next, highlight the main contribution of the research and propose future directions that stem logically from the current findings. A compelling conclusion often includes a brief call to action—whether for further inquiry, clinical translation, or policy consideration—based on the study’s outcomes. Importantly, the conclusion should not only showcase the authors’ expertise and critical insight but also acknowledge unresolved questions, theoretical limitations, or methodological challenges. Framing the findings within the larger scientific landscape reinforces the relevance of the work and underscores the ongoing need for research in this domain.
The manuscript contains no figures, five tables, and 114 references. This study stands out by integrating risk stratification into multimodal physiotherapy for children with chronic pain. It presents a fresh and practical approach. The interventions are not just described; they are carefully personalized. That is powerful. The Pediatric Pain Screening Tool provides a structured and adaptable method to tailor care. Results show meaningful clinical changes, not just statistical ones. With clear improvements in pain and psychosocial outcomes, this research adds value to pediatric pain management and supports individualized care strategies. I hope that after careful revision, the manuscript meets the journal’s high standards for publication. In addition, I anticipate the authors preparing “a detailed point-point rebuttal” to my remarks.
I declare no conflict of interest regarding this manuscript.
Best regards,
Reviewer
Author Response
Comment 1- Title: Provide a concise and informative title that accurately reflects the key message of this study, as this is the most essential aspect of the manuscript. Suggestions: Stratify, Specify, Satisfy: Multimodal Physiotherapy for Pediatric Chronic Pain; Precision in Motion, Protection from Emotion: Risk-Guided Therapy in Youth Pain; From Screening to Strength: Tailored Physiotherapy for Pediatric Pain Relief.
Response: Thank you for your comment. The title has been changed.
Comment 2- Abstract: Please structure the abstract by summarizing the background, methods, results, and conclusion in a well-balanced 200-word passage, without using section headings. Begin with a concise general introduction (1–2 sentences), followed by a more specific context (2–3 sentences) that leads into the research problem. Transition smoothly into the study’s objectives. When presenting results, focus on the key findings clearly and concisely, ending with a sentence that connects them to the wider scientific discussion. The conclusion should open with a strong statement, such as "This study demonstrates," emphasizing its main contribution. Wrap up with 2–3 sentences that provide a broader perspective, making the findings relevant and engaging for a diverse scientific audience. Avoid presenting statistical details in the abstract.
Response: Thank you very much for your suggestions, we think these help to improve the quality of the manuscript. The entire abstract has been restructured according to your comments. A 244-word abstract has been implemented, with introduction, gap in literature, objective of the study, methods, results and conclusions, also we included the p-values and the effect size.
Comment 3- Keywords: Make sure the majority of keywords are from Medical Subject Headings (MeSH) and use as many keywords as possible in the title and the first two sentences of the abstract.
Response: Thank you very much for your comments. We change some key word in other to include the most number of MeSH terms.
Comment 4- Graphical abstract: I highly recommend presenting an informative graphical or video abstract.
Response: Thank you very much for your comment, we included a video abstract in other to improve the quality and the impact of this study.
Comment 5- Introduction: Expand this section to improve clarity and reader engagement. Begin with a broad context to set the stage, then gradually narrow the focus to the specific issue being examined. Organize the introduction into well-structured paragraphs, aiming for a total of around 1000 words to thoroughly develop key concepts. Ensure accessibility for readers from different disciplines by clearly explaining the study’s purpose. Move logically from general background to a precise statement of the research objectives. Highlight the study’s significance by emphasizing how it addresses gaps in the field. Maintain a smooth, coherent flow to guide the reader seamlessly into the next sections, making the introduction both comprehensive and compelling. Avoid numbering in the text.
Response: Thank you very much for your suggestions. The introduction has been expanded, considering your comments, and further deepening the gap in literature, as well as the future implications of this approach that we are now suggesting.
Comment 6- Methods: To enhance clarity and reproducibility, detail the therapists’ qualifications and standardization procedures used across sessions. Including a session fidelity checklist or protocol adherence rating would bolster methodological rigor. Short, structured descriptions of each intervention phase can also aid replication and provide clearer insight into clinical decision-making processes. Avoid using bullets; rather, consider presenting tables.
Response: Thank you for your helpful suggestions. The content of the intervention sessions is now described in Table 1, as indicated in the manuscript. We have also included a description of the physiotherapists' training and qualifications, as well as the procedures used to ensure treatment fidelity. Specifically, we now report that all participants completed the full intervention as planned, and a fidelity check confirmed adherence to the standardized protocol.
Comment 7- Results: Consider integrating visual summaries such as bar graphs or line charts to illustrate key pre- and post-intervention changes. These visuals can enhance reader comprehension. Additionally, briefly highlighting effect sizes or clinical significance alongside p-values would deepen interpretation. Keep numerical results concise while emphasizing the most meaningful trends across risk groups. Avoid including statistical details within the main text and instead direct readers to the tables for reference. To enhance clarity, wrap up this section with a concise summary that highlights the key findings. A well-structured final paragraph will provide a clear overview of the results while emphasizing their importance in the broader scope of the study.
Response: Thank you for these valuable suggestions. We have now included effect sizes for all outcome measures to support the interpretation of the magnitude of change, alongside p-values. This addition provides a clearer understanding of the clinical relevance of the results. Additionally, we have added a concluding paragraph that summarizes the most meaningful findings across risk groups and emphasizes their potential clinical relevance within the broader context of pediatric pain management. We line charts, all key pre- and post-intervention data which presented in the figure 3. In the results section, we have streamlined the numerical details and directed readers to the tables for specific values.
Comment 8- Discussion: This section is well written. Please refer to the following general guidelines for crafting a discussion section. I recommend not presenting subheadings in the discussion. Present the discussion as a unified narrative, avoiding subsections, and organize it into several paragraphs totaling around 1,500 words. Start with an opening paragraph that introduces the discussion, and end with a concise summary that ties back to the main findings from the results. Use this section to build arguments that address the study’s primary objectives, key challenges, and the advancements in knowledge or tools needed to tackle them. Place the findings within a broader field perspective, emphasizing the significance of this research and showcasing how it expands on prior work while offering new insights. Discuss the theoretical and practical implications, as well as how the study can inspire future research. To enhance the discussion, evaluate the strengths and limitations of the study and reflect on its relevance to clinical applications, highlighting the practical impact of the findings. This structure will ensure the discussion is balanced, engaging, and meaningful.
Response:Thank you very much for your constructive comment. Your feedback has been very helpful in improving the quality of the manuscript. While we appreciate your recommendation to present the discussion as a unified narrative, we have decided to retain the subheadings to facilitate clarity and guide the reader through the different aspects of our analysis. Given the multidimensional nature of the intervention and the clinical context, we believe that a structured discussion allows for better organization and accessibility. Nonetheless, we have ensured that the discussion flows coherently and concludes with a concise summary that connects back to the main findings and objectives of the study.
Comment 9- Conclusion: This section is also well written. Just for the sake of improvement, I would appreciate it if the authors could benefit from the following general guidelines: The conclusion section should be thoughtfully crafted to deepen the reader’s understanding of the study’s significance with approximately 150–200 words. Ideally, this section should begin with a concise summary of the authors’ key message, followed by a clear presentation of the key research findings. Authors should then contextualize these results by discussing their broader theoretical and practical implications, emphasizing how the study advances knowledge in the field. Next, highlight the main contribution of the research and propose future directions that stem logically from the current findings. A compelling conclusion often includes a brief call to action—whether for further inquiry, clinical translation, or policy consideration—based on the study’s outcomes. Importantly, the conclusion should not only showcase the authors’ expertise and critical insight but also acknowledge unresolved questions, theoretical limitations, or methodological challenges. Framing the findings within the larger scientific landscape reinforces the relevance of the work and underscores the ongoing need for research in this domain.
Response: Thank you very much for your comments. Some sentences have been included in the conclusions section to emphasize the advances knowledge in pediatric pain field and including a call to action for stakeholder to support a real-world implementation of risk-stratified multimodal physiotherapy
Round 2
Reviewer 1 Report
Comments and Suggestions for Authors
Dear Authors,
Thank you very much for allowing me to review the manuscript again. The authors have adequately addressed all the issues raised.
Reviewer 3 Report
Comments and Suggestions for Authors
9 May 2025
The 2nd review on the manuscript, titled ‘Multimodal physiotherapist intervention for physical and psychological functioning in children with chronic pain: in-sights from risk stratification with the Pediatric Pain Screening Tool and recommendation to clinical practice’ by Ceniza-Bordallo M, submitted to Journal of Clinical Medicine
Dear Authors,
I appreciate the authors' thorough responses to the concerns raised in the previous review round. The revised manuscript is now a well-structured and clearly written research article that explores the effectiveness of a multimodal physiotherapy intervention tailored by Pediatric Pain Screening Tool (PPST) risk stratification to improve physical and psychological outcomes in children with chronic pain. This work contributes to validating these proteins as potential biomarkers for Parkinson’s disease with cognitive impairment. The manuscript upholds the high standards expected by the journal, and I look forward to future publications from these authors.
Thank you.
I declare no conflict of interest regarding this manuscript.
Best regards,
Reviewer